# R2C: Mapping Room to Chessboard to Unlock LLM As Low-Level Action Planner

## Abstract

This paper explores the potential of leveraging large language models (LLMs) as low-level action planners capable of executing long-horizon tasks based on natural language instructions. Although LLMs can act as the "brain" of robots by excelling in high-level task planning, they are not yet capable of directly guiding the "body" to execute low-level motion plans. This limitation stems from a communication gap between the "brain" and the "body". Specifically, LLMs lack access to rich spatial semantic information from the robot's real-time observations, hindering their ability to generate precise and actionable low-level plans. To address this, we propose a unified framework that bridges high-level and low-level planning by establishing an efficient communication interface between LLMs and robots. Our insight is to formulate the task as playing chess with LLMs. We map the room into a semantic chessboard, which we call **R**oom **to** **C**hessboard (**R2C**). Each grid represents the position and size of objects inside the room. We find that chessboard is **succinct** enough for LLMs to conduct semantic searches with global view of the room. Also, the chessboard is **informative** enough to convey detailed environmental state for LLMs to predict executable low-level actions. Additionally, we enhance decision-making through a Chain-of-Thought (CoT) paradigm, improving LLMs' interpretability and action reasoning. We implement R2C using both fine-tuned open-source LLMs and closed-source models like GPT-4, and demonstrate its efficacy on the challenging ALFRED benchmark. Our results show that with communication based on chessboard, LLMs can serve as effective low-level action planners, and can generalizes well to different settings and open-vocabulary robotic planning tasks. View the demos on our project page: https://anonymous4cv.github.io/Room2Chessboard/.

## 1 Introduction

The pursuit of general embodied agents focuses on developing robust systems that capable of understanding natural language commands to meet a wide range of human requirements. Traditional robotic learning methods have shown success in executing complex embodied tasks. However, they often face difficulties in generalizing to unseen environments or novel tasks due to their dependence on task-specific training data and rigid planning mechanisms. Recently, Large Language Models (LLMs) have emerged as promising candidates for enhancing embodied agents. Leveraging vast amounts of training data, LLMs exhibit strong generalization abilities across various domains. This makes them particularly suitable for tasks requiring flexible reasoning and decision-making in complex, free-form settings.

Several pioneering works Huang et al. (2022a); Driess et al. (2023); Song et al. (2023); Huang et al. (2022b) have explored the potential of LLMs as the "brain" of the embodied systems. Their strong generalization enables them to apply knowledge across diverse scenarios, effectively managing a wide range of embodied tasks. However, most existing LLM-based agents Huang et al. (2022a); Ahn et al. (2022); Song et al. (2023) primarily focus on high-level task planning, where LLMs decompose long-horizon tasks, e.g., "bring me an egg" into subgoals such as "go to egg" → "take it back". Translating these subgoals into executable low-level actions still *delegated to* APIs such as low-level policy networks Chaplot et al. (2020); Jang et al. (2022); Kalashnikov et al. (2021) trained on robotic trajectory data or deterministic algorithms like Sethian (1999); Dijkstra (2022). Nevertheless, while LLMs possess extensive world knowledge, they lack the spatial awareness of

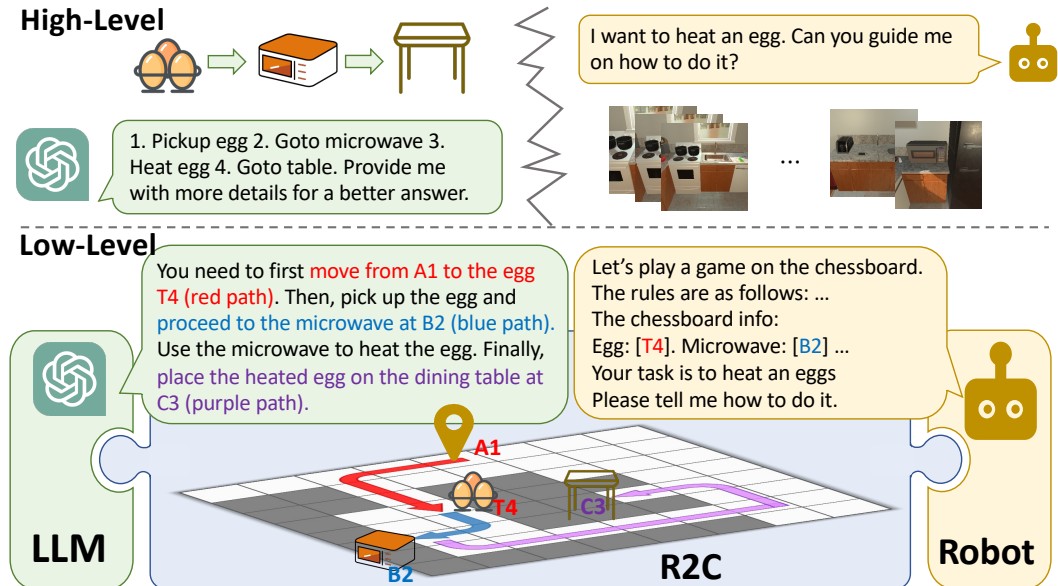

Figure 1: Comparison of LLM as a high-level planner versus low-level planner for the task of "heating an egg". While most existing methods use LLMs only for high-level task decomposition (top), the R2C framework converts embodied tasks into a chess-like game (bottom). This enables the LLM to perform executable low-level planning, guiding the robot through the path "A1-T4-B2-C3" to complete the task.

real-world environments. This limitation makes it difficult for LLMs to accurately predict the affordances and feasibility of specific actions. This often necessitates re-planning by the LLM Ahn et al. (2022); Song et al. (2023), which can reduce the overall efficiency of these frameworks in real-world applications and may impact task success rates under complex conditions. Hence, LLM's powerful reasoning capabilities are constrained.

The core barrier Song et al. (2023); Yang et al. (2023) to unleashing the potential of LLMs lies in the ineffective communication between the LLM (brain) and the Robot (body). On one hand, the Robot can hardly convey spatial information from the environment to the LLM, leaving the LLM devoid of decision-making grounds. On the other hand, the LLM struggles to efficiently communicate low-level decisions to the Robot, resulting in the Robot relying solely on external navigation APIs. Therefore, there is a necessity to establish a "common language" between LLM and the Robot — A platform to deliver sufficient environment information and low-level plans.

We propose a novel **R**oom **to C**hessboard (R2C) framework, which transforms complex embodied planning tasks into a chess-like game. As shown in Fig.1, R2C utilizes a grid-based chessboard as an effective communication platform between the robot and the LLM. On one side, the robot can continuously maps its observations onto the chessboard, which contains essential information such as object semantics, scene layout, and object size, while avoiding information overload. On the other side, the LLM is unlocked as a low-level action planner, guiding the robot through explainable "moves" to complete the task.

Specifically, R2C realizes high-level and low-level planning in a unified framework. The LLM first decomposes a long-horizon task into a sequence of subgoals. Each subgoal is then tackled during the low-level planning stage. At this stage, we introduce an Environment Filter to maintain the task-aware environment information onto a compact chessboard. Since the chessboard grid size is calibrated to match the robot's step length, the LLM can perform low-level planning on the chessboard by predicting the robot's next position. To further enhance LLM's understanding of the game, we formalize a **C**hain **o**f **T**hought **D**ecision (CoT-D) task for LLM to enhance its overall spatial reasoning and decision-making capabilities.

To construct a chessboard representation that aligns with the complexity of the environment while remaining manageable for LLMs, we design an Environment Filter to abstract the environment. Initially, new observations are transformed into a detailed 3D semantic map. However, such high-

dimensional representation can overwhelm LLMs. To address this, we apply kernel filters to down-sample the map, and then flatten it into a 2D chessboard based on the current subgoal. The object occupancy on the chessboard is further abstracted into coordinate sets. Such chessboard strikes a balance between semantic richness and simplicity, capturing essential information such as scene layout and object size, while enabling effective low-level action planning.

In R2C framework, LLM needs robust long-context understanding capabilities to comprehend the object coordinate sets and spatial reasoning abilities for low-level planning. Despite advancements in LLM capabilities, achieving this remains a huge challenge. To enhance LLM's low-level action planning capacity, we design a fine-tuning paradigm and formalized the CoT-D fine-tuning task. CoT-D comprises four subtasks: key information extraction, direction determination, target prediction, and selection analysis. LLM is required to link these subtasks together into a coherent logical chain Wei et al. (2022). This task can not only strengthen the long-context understanding of LLM but also enhance the spatial reasoning of model to generate more interpretable low-level plans.

We test our R2C framework on the comprehensive ALFRED Shridhar et al. (2020) benchmark, which features a diverse set of challenging long-horizon tasks. We test both zero-shot GPT prompts and fine-tune open-source LLMs using our novel CoT-D tasks. R2C achieves state-of-the-art (SoTA) performance among LLM-based methods. Additionally, LLMs trained with CoT-D show strong spatial perception and task-planning capabilities. Moreover, our low-level action planner demonstrates strong generalization across open-vocabulary tasks, showcasing the versatility and robustness of the R2C framework in complex environments.

The main contributions of this paper include:

1) Propose Room to Chessboard (R2C), an efficient communication platform between LLM and Robot to unlock LLM as low-level action planner.

2) Develop an explainable chain of thought decision analysis paradigm to guide LLMs make more reasonable and efficient robotic planning.

3) Achieve SoTA performances among LLM-based methods on complex long-horizon robot planning tasks using limited robotic data and show strong generalization to open-vocabulary tasks.

## 2 RELATED WORKS

### 2.1 TASK PLANNING IN ROBOTICS

Task planning in robotics Shridhar et al. (2020); Puig et al. (2018) involves generating a sequence of actions for robots to execute in the environment to achieve a specific goal. In real-world applications, the instructions are typically complex, resulting in long-horizon tasks that encompass a variety of embodied activities, such as navigation Anderson et al. (2018a); Gu et al. (2022) and object interaction Levine et al. (2018).

Early approaches Pashevich et al. (2021); Suglia et al. (2021) simply integrate visual and textual inputs to generate contextually appropriate action sequences. Besides, one widely used approach is reinforcement learning (RL) Majumdar et al. (2020); Wang et al. (2019); Tan & Bansal (2019); Hong et al. (2020). Although the above end-to-end methods have good performance, they rely entirely on instruction-trajectory training data, resulting in poor generalization to unknown environment Min et al. (2021); Kim et al. (2023); Song et al. (2023).

Therefore, some recent approaches Min et al. (2021); Blukis et al. (2022); Kim et al. (2023); Inoue & Ohashi (2022) break down the complex task into the multiple modules to reduce the high data cost. FILM Min et al. (2021) propose a framework with four submodules including language processing, semantic mapping, semantic search, deterministic policy. However, they rely on task prior like template-based instruction analysis and try using deterministic or a pre-trained low-level controller, which make them struggle in adapting to novel robotic tasks.

### 2.2 TASK PLANNING WITH LLMS

To address these challenges, incorporating Large Language Models (LLMs) into robotic task planning presents a promising pathway toward more adaptable robotic systems Huang et al. (2022a); Ahn et al. (2022); Rana et al. (2023); Huang et al. (2022b); Gu et al. (2023). Early attempts Huang et al.

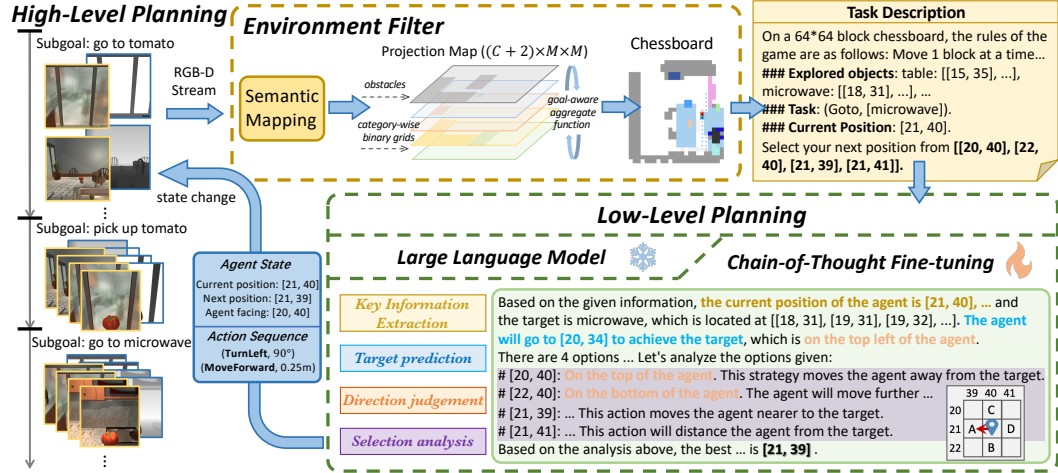

Figure 2: Overview of R2C: After *high-level planning*, the *nvironment filter* converts the current environment state into a grid-based chessboard. The state, along with game rules, forms the prompt input for the zero-shot or fine-tuned LLM. Through interpretable CoT decision analysis, the LLM predicts the next position, which the robot executes, creating a loop with updated visual information.

(2022a) adopt LLMs to help with free-form human instruction following by decompose the task into reasonable subgoals. However, due to LLM's inability to perceive the complex real physical world, the generated goals often fails to execute in the environment Ahn et al. (2022).

To address this, recent research has delved into integrating environment state information to ground the output of the LLM-based planner. LM-Nav Shah et al. (2023) additionally utilize a pre-trained Vision-Language Model (VLM) to generate visual captions and grounding the textual landmarks in the topological map to enhance the executable of navigation plan. The object-centric scene graph or topology graph is also utilized to enhance the LLM's understanding of the relations among objects within the environment Gu et al. (2023); Zhou et al. (2024); Chen et al. (2024). Other than directly translate the visual feedback from the environment to LLM, Say-Can Ahn et al. (2022) trained a vision-based value functions to judge the affordance of the LLM's plan. Some other works Liu et al. (2023) involve text-form Planning Domain Definition Language (PDDL) descriptions of a scene, which are more easier for LLM to handle.

Although the above work has solved some cases where LLM plans are not executable, they all use LLM as a high-level planner. Most works still relies on low-level controller to translate these high-level plan into executable action sequences. Some recently works Huang et al. (2024); **?** try using end-to-end framework. Different from them, we address this challenge by introducing an more interpretable pipeline. We map the room to chessboard to enable LLM directly make low-level plans according to the updated environment state.

## 2.3 SCENE REPRESENTATIONS

Scene representation is crucial for embodied tasks, particularly when using LLMs as planners. Topology graphs Anderson et al. (2018b); Ku et al. (2020) are often used for navigation but limit movement to predefined node connections, reducing their practicality in real-world applications. Object-centric scene graphs Gu et al. (2023); Rana et al. (2023) offer object-relative positions but lack the spatial details needed for low-level motion planning.

In contrast, our proposed chessboard representation incorporates both physical spatial and semantic information, making it suitable for a wide range of navigation tasks. As an abstraction of the semantic map Min et al. (2021); Lu et al. (2023), this grid-based chessboard balances conciseness with semantic richness, preserving essential details like object semantics, scene layout, and size, while avoiding the complexity of high-dimensional data.

## 3 METHODS

We aim to unlock LLMs as low-level action planner that can follow the instructions to solve long-horizon tasks. We first introduce our newly designed Room to Chessboard framework in Sec.3.1, and then illustrate the Chain-of-Though Decision (CoT-D) fine-tuning task formulation in Sec.3.2.

### 3.1 ROOM TO CHESSBOARD

Previous works, such as LLM-Planner Song et al. (2023) and SayCan Ahn et al. (2022), rely solely on LLMs to parse instructions and decompose tasks into high-level subgoals. The execution of each subgoal is then handled by a predefined controller that generates a sequence of low-level actions. In contrast, we introduce the Room to Chessboard (R2C) framework (Fig. 2), which enables LLMs to perform both High-Level Planning (HLP) and Low-Level Planning (LLP) within a unified framework.

First, the LLM conducts HLP, processing step-by-step instructions into a sequence of subgoals. Each subgoal is then handled at the LLP stage. At each timestep, an egocentric RGB-D image is processed by an Environment Filter to generate a chessboard abstraction. The updated chessboard is subsequently converted into object occupancy coordinates, which are fed to the LLM. Acting like a chess player, the LLM predicts the robot's next move on the chessboard, guiding it to navigate or interact with the environment.

#### 3.1.1 HIGH-LEVEL PLANNING (HLP)

At the beginning of an episode, the LLM processes the natural language instruction $L$ into a sequence of high-level subgoals, denoted as $\mathcal{G} = [G_1, G_2, ..., G_K]$. Each subgoal $G_k$ is a tuple (Action, Object), where Action $\in A_H$ is a primitive action chosen from the set of navigation action (GOTO) or interaction actions (e.g., PICKUP, PUT). The Object refers to the semantic class of the interacted object (e.g., SAFE, CD). These subgoals are executed sequentially during the LLP stage.

#### 3.1.2 ENVIRONMENT FILTER

Since the environment is partially observed, the agent has never explored the entire environment and must construct a map based on its own observations online. At each timestep, the Environment Filter $F(o_t, G_k)$ takes the new observations $o_t$ as input to filter out the goal-related environment information conditioned on the current subgoal $G_k$. This filtered information is then updated on a dynamically updated semantic chessboard $\mathcal{B}$ to represent the current state of the environment. The computation of $F$ involves two parts: *semantic mapping* and *map to chessboard*.

***Semantic mapping.*** We first build an online semantic map of the room, inspired by prior work Min et al. (2021). At each timestep $t$, the agent receives an egocentric RGB-D observation $o_t = \{I_{rgb}, I_{dpt}\}$. An off-the-shelf instance segmentation model processes $I_{rgb}$, and combined with $I_{dpt}$, the observation is converted into a point cloud, with each point labeled with predicted semantic categories. This 3D point cloud is then voxelized and flattend along the Z-axis to form the 2D semantic map $\mathcal{M}$. The resulting semantic map is represented as an allocentric $(C + 2) \times M \times M$ binary grid, where $C$ is the number of object categories while two additional channels denote obstacles and explored areas in each cell. Here, $M$ represents the map resolution.

***Map to chessboard.*** We further abstract the map into a compact chessboard. *Map to Chessboard* (M2C) function consists of two cascaded kernels. The first kernel is a dilation kernel $\mathcal{K}_{dila}$, to prevent collisions between the agent and obstacles. Similar to the dilation algorithm in image processing Serra (1982), it expands all occupied pixels of obstacles (including objects) outward by $\delta$ in the map. The second kernel is a max pooling kernel $\mathcal{K}_{pool}$, which performs max pooling on the map. Then the map is down-sampled from size $M$ to size $W$, where $W = \lceil \frac{D}{\omega} \rceil$ and $\omega$ is the grid size, which is calibrated to match the length of agent's step.

Next, we aggregate this map with multiple object layer into a unified single-layer chessboard. To address the many-to-one projection problem, where overlapping objects might appear in the same grid, e.g., "apples on a table", we introduce a *Goal-aware Aggregate* function $\mathcal{A}(.)$. This function prioritizes both the relevance of the current subgoal, i.e., the target object Object in current subgoal $G_k$, and the size of the object. The most relevant and smallest objects are placed at the top layer,

and then we merge object layers from bottom to top to produce the final chessboard. The overall formulation of the M2C is:

$$\text{M2C} = \mathcal{A} \circ \mathcal{K}_{dila} \circ \mathcal{K}_{pool}$$
$$\mathcal{B} = \text{M2C}(\mathcal{M}, G_k). \tag{1}$$

### 3.1.3 Low-Level Planning (LLP)

The chessboard filtered out from the Environment Filter provides compact yet sufficient environmental state information for LLM. We then simulate a "chess game" between the LLM and the robot to perform low-level planning. The LLP task is formulated as a single-step generation task for the LLM. As shown in Fig. 2, various state information is collected, including current subgoal $G_k$, chessboard state $\mathcal{U}$, action history $\mathcal{Q}$ and game rules $R$. Our prompt system $\mathcal{P}$ organizes this data and feeds it into the LLM to generate the next position prediction $u$. Note that LLP is invoked only for navigation subgoals, as interaction subgoals in ALFRED can be handled by predefined low-level actions once the agent reaches the visible range of the target object.

**Chessboard state.** To translate the chessboard state for the LLM, we convert it into textual object occupancy coordinate sets. The chessboard coordinate system originates from the upper left corner, with the X-axis pointing down and the Y-axis to the right. For each object on the chessboard, we gather a set of occupancy coordinates $\mathcal{U}_t^c = \{x_i, y_i\}$, where $c \in [1, C+2]$ and $x_i, y_i \in [1, W]$. The agent's current position is the previous step's target position $w_{t-1}$, and the initial agent position $w_0$ is predefined in the benchmark.

**Game Rules.** To assist the LLM in planning, we define key game rules $R$: 1) Basic inputs, including chessboard dimensions $W$, maximum steps $T$, and maximum failures $F$; 2) Action space, simplified to adjacent grids in four directions (up, down, left, right), i.e., the robot is only allowed to move 1 block at a time, due to the limited spatial reasoning ability of current LLMs; 3) Collision rules, where grids occupied by objects or obstacles are considered illegal moves.

**Action history.** We maintain a queue $\mathcal{Q}$ with a fixed length $\tau$ to store recent successful movement coordinates, providing the LLM with contextual information about previous actions.

At each step $t$, the prompt system combines all this information and feeds it to the LLM, which simulates a chess player to decide the next move. The LLP models a policy $\pi$, defined as:

$$u_t = \pi(\mathcal{P}(G_k, \mathcal{U}_t, R, \mathcal{Q}_t)), \tag{2}$$

where $u_t = (x_t, y_t)$ is the predicted next position on the chessboard. According to the defined action space, the available next positions are restricted to the adjacent grids in four directions $x_{t+1} \in [x_t - 1, x_t + 1]$ and $y_{t+1} \in [y_t - 1, y_t + 1]$, with $x_{t+1}, y_{t+1} \in [1, W]$.

Since the chessboard grid size is calibrated to match the robot's step length, the predicted adjacency target position $u_t$ can be directly converted to a sequence of executable low-level actions $\mathbf{a} = \{a_i\}, a_i \in \mathbf{A}_{nav}$ in the real-world environment. For example, in the ALFRED benchmark, $\mathbf{A}_{nav}$ includes (MoveForward, $0.25m$), (TurnLeft, $90°$) and (TurnRight, $90°$). Based on the predicted target position, the movement in any of the four directions can be converted into the corresponding action sequence (e.g., TurnLeft and MoveForward to move to the left adjacent grid). Additionally, if the output $w_t$ is within the visible range of the target object OBJECT in the current subgoal, the system moves on to the next subgoal. See more details of the complete R2C planning pipeline in pseudo code Algo.1 in appendix.

### 3.2 Chain-of-Thought Fine-tuning Paradigm

To enhance the decision-making capabilities of LLMs in our chess game, we introduce an interpretable fine-tuning paradigm with two key features: 1) simultaneous training of high-level Task Decomposition (TD) and low-level Chain of Thought Decision (CoT-D) tasks to unify HLP and LLP in within a single LLM, and 2) a Chain-of-thought Decision (CoT-D) process composed of four sub-parts are designed to enhance LLM's rule comprehension and spatial reasoning abilities.

### 3.2.1 TASK DECOMPOSITION (TD)

We leverage the exceptional natural language understanding capabilities of LLMs to interpret and follow instructions. Especially when utilized to effectively decompose complex, long-horizon tasks into manageable subgoals. For the input instruction description $L$, it will be decomposed into a sequence of sub-steps $\{G_1, G_2, ..., G_K\}$, where each sub-step is a tuple of action and target object, [Action, Object].

### 3.2.2 CHAIN OF THOUGHT DECISION (COT-D)

LLM must thoroughly comprehend our chessboard coordinate system and predict the next position based on the semantic information of the chessboard and its inherent common sense. For the given prompt $\mathcal{P}$ with chessboard state $\mathcal{U}$, the LLM needs to generate answer sentence $\mathcal{S}$ (including next position $w$) with probabilistic language model $p_{LM}$. If LLM is asked to directly provide coordinates, this process can be formalized as:

$$p(\mathcal{S} \mid \mathcal{P}) = \prod_{i=1}^{|\mathcal{S}|} p_{LM}\left(s_i \mid \mathcal{P}, s_{<i}\right) \qquad (3)$$

However, such a complex task is much more challenging than the task decomposition. This requires that LLMs have strong capabilities in long-text comprehension and spatial reasoning. However, the current leading models are not particularly skilled in these areas. Consequently, we design the Chain of Thought Decision (CoT-D) tasks to strengthen LLM's rationale $\mathcal{R}$ in these aspects. The task consists of four sub-parts, each requiring LLMs to output the result of key information extraction $\mathcal{R}_E$, direction judgment $\mathcal{R}_D$, target prediction $\mathcal{R}_T$, and selection analysis $\mathcal{R}_S$. We link these sub-tasks sequentially in natural language to construct a coherent logical chain. The entire task can be represented as:

$$p(\mathcal{S} \mid \mathcal{P}) = p(\mathcal{S} \mid \mathcal{P}, \mathcal{R}) \cdot p(\mathcal{R} \mid \mathcal{P})$$
$$p(\mathcal{R} \mid \mathcal{P}) = \prod_{r_i \in \{\mathcal{R}_E, \mathcal{R}_D, \mathcal{R}_T, \mathcal{R}_S\}} p_{LM}\left(r_i \mid \mathcal{P}, r_{<i}\right)$$
$$p(\mathcal{S} \mid \mathcal{P}, \mathcal{R}) = \prod_{j=1}^{|\mathcal{S}|} p_{LM}\left(s_i \mid \mathcal{P}, \mathcal{R}, s_{<j}\right) \qquad (4)$$

The specific content of the four sub-tasks will be introduced below. Examples can be found in Fig.2, with details available in the appendix.

***Key Information Extraction:*** Due to the intricate information contained within the chessboard, when textualized, it becomes a lengthy document with substantial information. Considering that LLM often encounters issues such as context loss when comprehending long texts, this task is designed to train LLM in processing task-related long texts and extracting key information. This task requires LLM to extract relevant information about the current coordinates and target objects based on the chessboard information inputs. The form used in data annotation is as follows: The current position of the agent is [COORDS], and the target is [OBJECT NAME], which is located at [COORDS SET].

***Direction Judgment:*** The conversion from a chessboard grid image to text is based on a two-dimensional Cartesian coordinate system, and all spatial relationship understanding relies on a good coordinate system understanding. Despite explaining the rationale for establishing this coordinate system in the instruction using prompts like Establish a coordinate system with the top left grid as (1,1), testing has shown that LLM still frequently misunderstands our settings. This could be attributed to the length of the text and the scarcity of spatial reasoning tasks incorporated during the pre-training of contemporary LLM. To ensure LLM's accurate comprehension of the chessboard grid coordinate system, this task requires LLM to judge the direction between the target and the current coordinates. The format employed in data annotation is as follows: the target is at..., which is on the [DIRECTION] of the agent.

***Target Prediction:*** The locations where different categories of objects appear often have priors. For example, sofas are likely to appear opposite the television. Therefore, we aim for LLM to develop the capability of predicting target locations. This can minimize the ineffective or inefficient exploration process for the robot to find target objects, thereby improving the efficiency of the system

in completing embodied tasks. The expression utilized in data annotation is: `Based on the chessboard analysis, the target is likely located near [COORDS]`.

***Selection Analysis:*** The tasks outlined above will enhance LLM's comprehension of the spatial relationships within the chessboard and the goals of the overall embodied task. Based on this, we require LLM to analyze all possible next positions according to the rules of the chessboard. Throughout this analysis process, utilizing the aforementioned analysis as a foundation, LLM will provide analysis for each choice. The specific expression in the annotation is: `[COORDS]: This position is on the [DIRECTION] of the agent, [Reason]`. The `[Reason]` part is derived from annotations generated by GPT-4 based on the agent's current state and the correlation between the choice and the correct orientation. For example: `Objects in this direction have already been discovered; we should head to areas that have not been explored`.

## 4 EVALUATION

### 4.1 EXPERIMENT SETTINGS

**Benchmark.** We evaluate our method on the challenging ALFRED Shridhar et al. (2020) benchmark, which includes 7 types of long-horizon tasks across 207 unique environments and 115 different object types. In our experiments, we adhere to the benchmark's settings, including low-level actions, maximum agent steps, and failure limits.

**Chessboard settings.** The physical room size $D$ is set to $16m$, which is set to be the approximate maximum room size, and the grid size $\omega$ is the length of the agent step, $0.25m$. Thus, the size of the semantic map and the chessboard is $M = 320$ and $W = 64$ separately. However, to represent more complex environment, one can choose a more fine-grained chessboard. We show such demo in the appendix6.3.

**Model settings.** For the zero-shot R2C, we use the public GPT-4-turbo API OpenAI (2023) without any example. Considering the testing cost, we randomly sampled a subset of size 100 covering all 7 types of tasks strictly according to the task distribution of ALFRED. For fine-tuning, we collected a total of 264,915 data samples for fine-tuning, including 70,000 samples for the Task Decomposition tasks and 194,915 samples for the Chain-of-Thought Decision tasks. All data collection was based solely on the training set of ALFRED. During training, we conduct full-parameter fine-tuning on both the Mistral-7B-Instruct-v0.2 model Jiang et al. (2023) and the Llama-7B-Chat model Touvron et al. (2023) using all the data. Both models were trained for only one epoch. We conduct all fine-tuning experiments using 4 NVIDIA H100 GPUs, and all evaluations are performed on 4 NVIDIA A40 GPUs. *The codes will be released after the paper is accepted.*

### 4.2 MAIN RESULTS

Tab.1 presents the evaluation results on the validation set of ALFRED. We compare our R2C framework with both traditional robotic learning methods (specialists) and LLM-based approaches (generalists). R2C achieves state-of-the-art performance among LLM-based methods, with the Mistral-7B excelling in seen environments and GPT-4 in unseen environments.

Compared to SayCan Ahn et al. (2022) and LLM-P Song et al. (2023), which use LLMs only for high-level planning, R2C integrates high-level and low-level planning, offering a more efficient, end-to-end solution. LLM-P requires 100 instruction-plan pairs for training, while R2C operates in a zero-shot setting without examples. Our CoT-D paradigm enables GPT to analyze the chessboard state comprehensively, improving decision interpretability and efficiency.

R2C fine-tuned on open-sourced LLMs achieves competitive results, with a 2.31% improvement in the seen split and a 1.65% drop in the unseen split compared to R2C-GPT-4. This highlights GPT-4's superior generalization to unseen environments. However, with our carefully designed fine-tuning tasks, R2C implemented on much smaller models like LLaMA and Mistral, can still deliver performance comparable to GPT-4. This demonstrates the effectiveness and efficiency of our fine-tuning approach.

Finally, R2C fine-tuned on collected data performs comparably to specialist models like FILM Min et al. (2021) and LEBP Liu et al. (2022). Although these models, specifically designed for ALFRED tasks, excel in seen scenes, they struggle in unseen environments due to overfitting. In contrast, R2C

Table 1: Main results on the ALFRED benchmark. SR and GC are short for success rate and goal-conditioned success rate. $\Delta$ SR and $\Delta$ GC represent the performance differences in generalizing from the seen to the unseen environment.

| Method | Training Mode | Valid Seen | | Valid Unseen | | $\Delta$ SR ↑ | $\Delta$ GC ↑ |
|---|---|---|---|---|---|---|---|
| | | SR ↑ | GC ↑ | SR ↑ | GC ↑ | | |
| *Specialists, only for ALFRED tasks* | | | | | | | |
| E.T. Pashevich et al. (2021) | from scratch | 46.59 | 52.92 | 7.32 | 20.87 | -39.27 | -32.05 |
| HiTUT Zhang & Chai (2021) | from scratch | 25.24 | 34.85 | 12.44 | 23.71 | -12.80 | -11.14 |
| M-TRACK Song et al. (2022) | from scratch | 26.70 | 33.21 | 17.29 | 28.98 | -9.41 | -4.23 |
| FILM Min et al. (2021) | from scratch | 24.63 | 37.20 | 20.10 | 32.45 | -4.53 | -4.75 |
| LEBP Liu et al. (2022) | from scratch | 27.63 | 35.76 | 22.36 | 29.58 | -5.27 | -6.18 |
| *Generalists, based on LLMs* | | | | | | | |
| SayCan Ahn et al. (2022) | few-shot | 12.30 | 24.52 | 9.88 | 22.54 | -2.42 | -1.98 |
| LLM-P (GPT) Song et al. (2023)[1] | few-shot | 16.45 | 30.11 | 15.36 | 29.88 | -1.09 | -0.23 |
| R2C-GPT-4 (ours) | zero-shot | 20.00 | 28.46 | **24.00** | 28.24 | **+4.00** | -0.22 |
| R2C-Llama-7B (ours) | fine-tune | 20.83 | 29.60 | 18.99 | 29.69 | -1.84 | **+0.09** |
| R2C-Mistral-7B (ours) | fine-tune | **22.31** | **32.40** | 22.35 | **31.97** | +0.04 | -0.43 |

Table 2: Ablation of R2C. GT Seg. represents the model using ground-truth segmentation. GT Goal represents using ground-truth subgoals. - SA is the model without Selection Analysis part of CoT. - CoT is the model without all CoT tasks.

| Method | Val Seen | | Val Unseen | |
|---|---|---|---|---|
| | SR | GC | SR | GC |
| Base Model | 22.31 | 32.40 | 22.35 | 31.97 |
| + GT Seg. | 37.92 | 45.83 | 35.24 | 43.88 |
| + GT Seg., GT Goal | **48.18** | **55.13** | **53.33** | **58.18** |
| + GT Seg., GT Goal, - SA | 45.97 | 51.75 | 47.45 | 53.03 |
| + GT Seg., GT Goal, - CoT | 41.22 | 49.35 | 41.96 | 47.88 |

Table 3: The performances of R2C on different tasks.

| Task Type | Val Seen | | Val Unseen | |
|---|---|---|---|---|
| | SR | GC | SR | GC |
| Overall | 48.18 | 55.13 | 53.33 | 58.18 |
| Examine | 75.86 | 79.31 | 83.33 | 87.96 |
| Pick & Place | 69.57 | 69.57 | 63.33 | 63.33 |
| Stack & Place | 29.03 | 37.00 | 45.45 | 46.46 |
| Clean & Place | 63.89 | 73.87 | 38.89 | 50.45 |
| Cool & Place | 36.84 | 49.60 | 61.11 | 69.23 |
| Heat & Place | 23.53 | 41.38 | 28.57 | 41.13 |
| Pick 2 & Place | 33.33 | 54.41 | 37.50 | 53.70 |

and other LLM-based methods demonstrate stronger generalization across both seen and unseen environments.

### 4.3 ABLATION STUDY

We performed an ablation study to analyze the impact of different R2C modules, as shown in Tab.2. Our base model uses raw RGB-D frames with pre-trained Mask-RCNN for instance segmentation, ensuring a fair comparison with previous works Pashevich et al. (2021), Min et al. (2021). Ground truth segmentation (GT Seg.) significantly improves performance, highlighting segmentation as a bottleneck, especially in simulation environments where domain gaps between simulated and real-world data are pronounced.

Using ground truth subgoals (GT Goal) also boosts SR, revealing the impact of ambiguity in natural language instructions. Annotator confusion between categories like desk lamps and floor lamps can hinder task decomposition, though LLMs' strong generalization helps mitigate this.

Finally, removing the CoT-D framework, which forces the model to directly output position coordinates without rationale, results in a significant performance drop, , especially in unseen scenes. This highlights the importance of fine-tuning with CoT-D subtasks for better spatial reasoning and action planning. Ablating individual subtasks is challenging due to their interdependence. We ablate removing the Selection Analysis part, where decisions are made without analyzing options and it causes a noticeable performance decline.

### 4.4 EVALUATION ACROSS TASKS TYPES.

We analyzed R2C's performance across task types using GT Segmentation and GT Goal settings, with task-specific results shown in Tab.3. The R2C model excels particularly in "Examine" tasks,

---

[1]For a fair comparison, we test the performance of the GPT-4 version LLM-P on the same selected subset as us, achieving SR of 16.21% on seen scenes. Since this work solely employs LLM for task decomposition, the performance is only significantly different from using GPT-3.

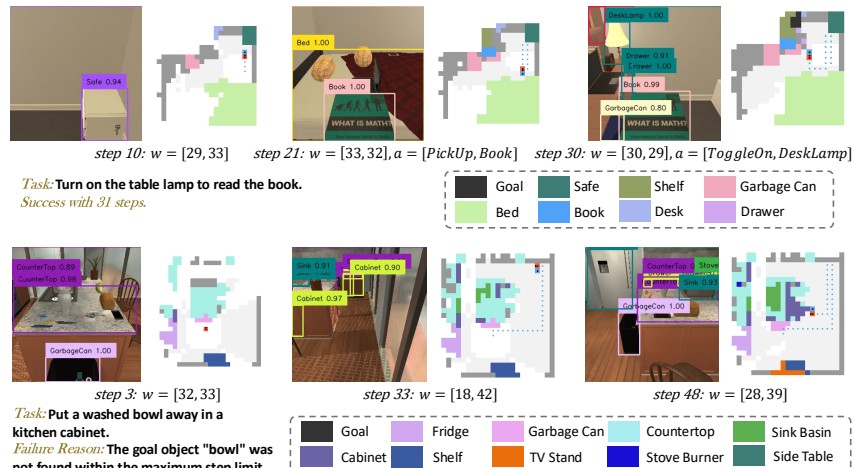

Figure 3: Cases study of our R2C with Mistral model. The model successfully finishes the task with 31 steps (up) while fails to find the small object in a complex environment (bottom).

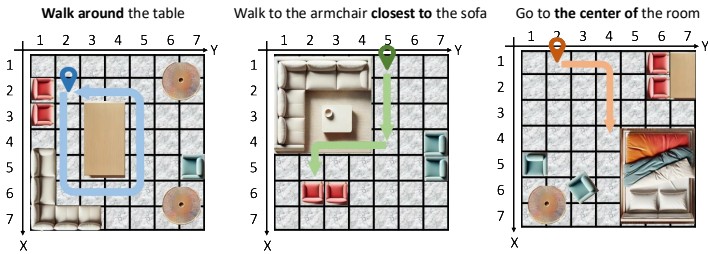

Figure 4: Cases of R2C-GPT-4 on open-vocabulary task. As the R2C framework unlocks LLM's low-level planning capability, LLM can tackle more open-ended tasks.

achieving the highest SR, but faces challenges in complex "Heat & Place" tasks, where the "HeatObject" subgoal alone involves seven interactive steps, increasing the risk of error accumulation. Besides, R2C shows strong navigation and exploration in "Pick & Place" tasks, benefiting from its direction judgment and goal prediction reasoning. We provide a visualization of two case study of R2C completing different tasks ("Examine" and "Clean & Place") in Fig.3.

### 4.5 EXPLORATION ON OPEN-VOCABULARY TASK

While tasks in datasets like ALFRED are predefined and limited to 7 templates with fixed execution orders, real-world demands are far more diverse. Standard instructions such as "walk around the table" render high-level LLM planners relying on navigation APIs ineffective, as there is no customized API for such tasks. We refer to these as open-vocabulary embodied tasks. In contrast, the R2C framework equips LLMs to directly make low-level action plans, enabling them to adaptively handle a wide range of tasks. As shown in Fig.4, we test open-vocabulary tasks with GPT-version R2C, demonstrating superior performance and highlighting the importance of integrating low-level planning for real-world applications.

## 5 CONCLUSION

We introduce the Room to Chessboard (R2C) framework, which map the complex room into a chessboard as a communication platform between the LLM and the robot, unlocking the LLM as low-level planner to directly guide the robot adaptively finish the embodied tasks. To address the spatial reasoning tasks on chessboard, we design a CoT fine-tuning framework that enhances LLMs' ability to make interpretable low-level decisions. Experiments show that R2C outperforms existing LLM-based high-level planners, even in zero-shot settings, and allows 7B models to surpass GPT-4. Furthermore, R2C enables LLMs to tackle open-vocabulary tasks where API-based frameworks fall short. However, R2C still faces challenges, such as handling very large scenes. Future work will focus on optimizing R2C for larger environments and exploring its potential for various open-vocabulary tasks.

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

# 6 APPENDIX

## 6.1 IMPLEMENTATION DETAILS OF CHAIN-OF-THOUGHT FINE-TUNING

### 6.1.1 MODEL PARAMETER SETTINGS

The hyperparameters for the Chain-of-Thought fine-tuning of Mistral-7B are detailed in Table 4. We employ Mistral-7B-Instruct-v0.2 as the base model, which is trained using a full fine-tuning approach. The learning rate is configured at 2e-5. For training, the batch size per device is 32, while during inference, it is reduced to 1.

| Hyperparameter | for Mistral-7B fine-tuning |
|---|---|
| base model | Mistral-7B-Instruct-v0.2 |
| fine-tune mode | full fine-tune |
| deep speed | ds_z3 |
| cutoff len | 3000 |
| preprocessing num workers | 16 |
| per device train batch size | 32 |
| per device eval batch size | 1 |
| gradient accumulation steps | 1 |
| lr scheduler type | cosine |
| logging steps | 10 |
| warmup steps | 375 |
| save steps | 500 |
| eval steps | 500 |
| evaluation strategy | steps |
| learning rate | 2e-5 |
| num train epochs | 1.0 |
| max samples | 1000000 |
| val size | 0.01 |
| ddp timeout | 1800000 |

Table 4: Hyperparameters in Chain-of-Thought fine-tuning of Mistral-7B.

## 6.2 COMPARISON OF THE COMPUTATION COST

| Model | FILM | LLM-P (GPT) | R2C-GPT-4 | R2C-Mistral-7b |
|---|---|---|---|---|
| Input Prompt Length (tokens) | / | ~1k | ~1k | ~1k |
| Output Seq. Length (tokens) | / | ~20 | ~1.5k | ~1.5k |
| Training Data / Examples (ep) | 21k | 100 | 0 | 21k |
| Training Time (GPU*Day) | 1.5[3090] | / | / | 4[A100] |
| Inference Speed (s/times) | 357[3090] | 5 | 12 | 0.53 (100 processes) / 3.79 (1 process) |
| Inference Freq. (times per ep) | 1 | <10 | ~50 | ~50 |

Table 5: Comparison of different models and configurations.

We provide the comparison of the computational costs among the R2C models, traditional baseline FILM, and LLM-based baseline, LLM-P. Note that inference speed refers to the time required to call the model each time, and inference frequency refers to the average frequency of calling the model per episode.

As shown in Tab.5, though the text results of the chessboard are relatively long, we only keep the task-related object coordinates and remove the unexplored coordinates. The prompt length of R2C in each turn is about 1k tokens, which is similar to the LLM-P. The proposed CoT-D mainly affects the length of the output of LLMs. Therefore, the inference time will become relatively longer. However, since LLM-P can only generate high-level plans, besides the inference time of GPT, its inference time depends on the motion planning model, HLSM, which is not short either, see the inference speed of FILM (357s).

Besides, the training data we use is similar to FILM. We just split the complete episode data into single steps to train the model's single-step prediction ability. Since the number of parameters is much larger than traditional models, the training time of R2C is greater than FILM. However, FILM has 4 sub-models to train. The combination of each sub-models will also bring significant time costs.

As for the inference cost, the average inference time of FILM is about 357 seconds for each episode. Our R2C-Mistral-7b completes the tasks through single-step reasoning. the inference time costing of an episode between FILM and R2C-Mistral-7b is on the same scale (357s vs. 8*50=400s). Note that the model and the task can be run in parallel using the vLLM speed-up technique (multi-process) to realize much faster inference. Using 4 A40 GPUs and run tasks in parallel (100 process) can speed up the single step inference time to 0.53s.

## 6.3 DETAILS OF ROOM TO CHESSBOARD FRAMEWORK

The pseudo code of the complete R2C pipeline is described in Algo. 1:

---
**Algorithm 1** Room to Chessboard (R2C) Planning Pipeline

---
1: **Input:** Instruction $L$, Game rules $R$, Initial chessboard information $\mathcal{U}_0$, Initial position $w_0$
2: **Output:** Sequence of coordinates $[w_t = (x_t, y_t)]$ and action sequence $\mathbf{a}$
3: Initialize current position $(x_0, y_0)$ based on $w_0$
4: Initialize history steps $\mathcal{Q}_0 = []$
5: $t \leftarrow 0, k \leftarrow 1, failures \leftarrow 0$
6: **High-Level Planning (HLP):**
7: Parse instruction $L$ to generate sequence of subgoals $\mathbf{G} = [G_1, G_2, ..., G_K]$, where $G_k = (Action_k, Object_k)$
8: **while** $t < T$ **and** $failures < F$ **and** $k \le K$ **do**
9:     Get current subgoal $G_k = (Action_k, Object_k)$
10:     **Low-Level Planning (LLP):**
11:     Get new observations: $o_t = \{I_{rgb}, I_{dpt}\}$
12:     Update chessboard: $\mathcal{U}_t = F(o_t, G_k)$
13:     **if** $Action_k$ is navigation **then**
14:         $w_t = \pi(P(G_k, \mathcal{U}_t, R, \mathcal{Q}_t))$           ▷ Predict next position
15:         Convert $w_t = (x_t, y_t)$ to executable actions $\mathbf{a} = \{a_1, a_2, ..., a_n\}$ in $\mathbf{A}_{nav}$
16:         Execute actions $\mathbf{a}$
17:         $t \leftarrow t + n$
18:         **if** Move to $(x_t, y_t)$ is successful **then**
19:             $\mathcal{Q}$.append($(x_t, y_t)$)
20:         **else**
21:             $failures \leftarrow failures + 1$
22:         **end if**
23:         **if** $Object_k$ is visible in $\mathcal{U}_t$ **then**
24:             $k \leftarrow k + 1$           ▷ Move to the next subgoal if the object is visible
25:         **end if**
26:     **else if** $Action_k$ is interaction **then**
27:         Execute predefined low-level interaction actions $\mathbf{a} = \{a_1, a_2, ..., a_n\}$ in $\mathbf{A}_{int}$
28:         $t \leftarrow t + n$
29:         $k \leftarrow k + 1$           ▷ Directly move to the next subgoal after interaction
30:     **end if**
31: **end while**

---

## 6.4 CASE STUDY OF MORE FINE-GRAINED CHESSBOARD

We further conduct experiments with the grid's size set to be $0.1m$. Then the chessboard becomes more fine-grained ($160 \times 160$). We use the R2C-Mistral-7b model trained on $64 \times 64$ chessboard data to infer with this more fine-grained chessboard (zero-shot). We show the visualization results in the Fig.5.

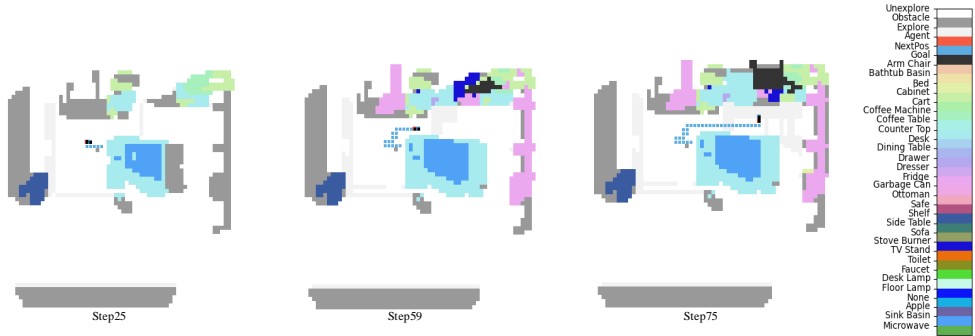

Figure 5: Cluttered setting demo. The chessboard size is $160 \times 160$. The task is "Get an apple from the sink and heat it up in the microwave". The blue dashed line represents the trajectory.

## 6.5 IMPLEMENTATION DETAILS OF GPT-4 EXPERIMENTS

In all our GPT-4 experiments on the ALFRED benchmark, we use the official API without any in-context examples. The specific model employed is gpt-4-turbo-2024-04-09.

For subgoal decomposition, we approach this as a translation problem. The central concept involves translating noisy high-level plan annotations into a subgoal format, structured as a subgoal verb with corresponding subgoal objects. Each high-level plan annotation is directly mapped to a subgoal expression.

In the prompt, we initially inform GPT-4 of the format and limitations of the subgoal expression, specifically the subgoal verb and objects. The subgoal verb is restricted to a predefined set: [GotoLocation, PickupObject, PutObject, CoolObject, HeatObject, CleanObject, SliceObject, ToggleObject]. For specific verbs like HeatObject, we clarify their usage. The subgoal objects are task-dependent; for each task, we maintain a list of relevant large and small objects, and GPT-4 is restricted to using only these listed items to predict the subgoal objects. We also provide GPT-4 with the task description and the high-level plan annotations in a list format, organized in chronological order of subgoals.

Finally, we define the output format for GPT-4 using a static example, employing a "—" to separate subgoal sequences. Upon receiving GPT-4's response, we process the reply to extract the subgoal list. This involves using the "—" delimiter in the output format to segment the sequence into a list, followed by subgoal cleaning. This cleaned list then guides the low-level actions.

For the prompts, we first ensure that GPT-4 understands the rules of a chessboard game, explaining the board size, coordinate system, and terms to describe spatial relations: left, right, up, down, leftup, rightup, leftdown, and rightdown. We then convert the current state of the chessboard into text, which includes object names and their occupied coordinates. We specify that GPT-4 cannot move to coordinates occupied by an object and can only travel one block at a time.

After setting up the chessboard game, we provide GPT-4 with all available information including the task description, current subgoal, current position, and available positions. The task description aligns with the current subgoal format. The candidates for available positions are the blocks immediately adjacent (up, down, left, right) to the current position, ensuring no objects occupy these blocks.

Finally, we instruct GPT-4 on its tasks and how to solve problems using a chain-of-thought approach. Specifically, we ask GPT-4 to verbalize its reasoning, analyzing the spatial relationship between the target and current positions, and between the current position and available positions. After reasoning, we prompt GPT-4 to indicate its next desired position and analyze whether the subgoal is achieved, thereby enhancing its understanding of the subgoals and the overall task. The output is formatted in JSON with keys for "reason", "next position", and "subgoal_done". Importantly, we

Please translate the subgoal instruction into subgoal expression like (GotoLocation,["Soapbar"]).
Each subgoal can be translated to ONLY ONE subgoal expression!For example, if you have 5 subgoal, you should only output 5 subgoal expression!
Please ONLY use the following subgoal verb:
['GoToLocation','PickupObject','PutObject','CoolObject','HeatObject','CleanObject','SliceObject','ToggleObject']
When the task means clean something, use CleanObject instead of ToggleObject something like faucet
When the task means heat or cook something, use HeatObject instead of PutObject into microwave etc.

Your task is Look at a mug in lamp light.
And the subgoal instruction are : ['Go to the desk in front of you, then look up to the shelves.', 'Pick up the mug from the shelf.', 'Look down to the lamp on the desk.', 'Turn on the lamp.']
Please ONLY use the following object name:
['Sink Basin', 'Arm Chair', 'Bathtub Basin', 'Bed', 'Cabinet', 'Cart', 'Coffee Machine', 'Coffee Table', 'Counter Top', 'Desk', 'Dining Table', 'Drawer', 'Dresser', 'Fridge', 'Garbage Can', 'Microwave', 'Ottoman', 'Safe', 'Shelf', 'Side Table', 'Sofa', 'Stove Burner', 'TV Stand', 'Toilet', 'Faucet', 'Floor Lamp', 'None', 'Mug', 'Desk Lamp']
ONLY answer with DIRECTLY like the example below , notice that an subgoal except PutObject only have one object and this example's task is "put a cooked slice of bread on to the counter":

(GotoLocation,["Countertop"])|(PickupObject,["Knife"])|(GotoLocation,["Bread"])|(SliceObject,["Bread"])|(GoToLocation,["Drawer"])|(PutObject,["Knife","Drawer"])|(GoToLocation,["Dining Table"])|(PickupObject,["Bread"])|(GoToLocation,["Countertop"])|(PutObject,["Bread","Microwave"])|("HeatObject",["Bread"])|(GoToLocation,["Countertop"])|(PutObject,["Bread","Countertop"])

**Answers:** GPT-4: (GotoLocation,["Shelf"])|(PickupObject,["Mug"])|(GotoLocation,["Desk Lamp"])|(ToggleObject,["Desk Lamp"])

Figure 6: GPT-4's high-level planning prompt and answer.

On a 64*64 block chessboard, the rules of the game are as follows:
Establish a coordinate system with the top left grid as (1,1). Each block can be represented by coordinates.
For instance, the block in the 3rd row and 5th column is denoted as (3,5). its right is (3,6), its left is (3,4), its up is (2,5), its down is (4,5).
spatial relation:[left,right,up,down,leftup,rightup,leftdown,rightdown]
In the chessboard, there are the following objects: Bed: [[14, 29], [14, 30], [14, 31], [14, 32], [14, 33], [14, 34], [14, 35], [14, 36], [14, 37], [14, 38], [14, 39], [14, 40], [14, 41], [14, 42], [14, 43], [15, 29], [15, 31], [15, 32], [15, 35], [15, 36], [15, 37], [15, 38], [15, 39], [15, 40], [15, 41], [15, 42], [15, 43], [16, 29], [16, 30], [16, 31], [16, 32], [16, 33], [16, 34], [16, 35], [16, 36], [16, 37], [16, 38], [16, 39], [17, 29], [17, 30], [17, 31], [17, 32], [17, 33], [17, 34], [17, 35], [17, 36], [17, 37], [17, 38], [17, 39], [18, 29], [18, 30], [18, 31], [18, 33], [18, 34], [18, 35], [18, 36], [18, 37], [18, 38], [18, 39], [19, 29], [19, 30], [19, 31], [19, 32], [19, 33], [19, 34], [19, 35], [19, 36], [19, 37], [19, 38], [19, 39], [20, 29], [20, 30], [20, 31], [20, 32], [20, 33], [20, 34], [20, 35], [20, 36], [20, 37], [20, 38], [20, 39], [21, 29], [21, 30], [21, 31], [21, 32], [21, 33], [21, 34], [21, 35], [21, 36], [21, 37], [21, 38], [21, 39], [22, 29], [22, 30], [22, 31], [22, 32], [22, 33], [22, 34], [22, 35], [22, 36], [22, 37], [22, 38], [22, 39], [23, 29], [23, 30], [23, 31]]
Unexplore: [[16, 40], [16, 41], [16, 42], [16, 43], [17, 40], [17, 41], [17, 42], [17, 43], [18, 40], [18, 41], [18, 42], [18, 43], [19, 40], [19, 41], [19, 42], [19, 43], [20, 40], [20, 41], [20, 42], [20, 43], [21, 40], [21, 41], [21, 42], [21, 43], [22, 40], [22, 41], [22, 42], [22, 43], [23, 40], [23, 41], [23, 42], [23, 43], [24, 39], [24, 40], [24, 41], [24, 42], [24, 43], [25, 39], [25, 40], [25, 41], [25, 42], [25, 43], [26, 38], [26, 39], [26, 40], [26, 41], [26, 42], [26, 43], [27, 38], [27, 41], [27, 42], [27, 43], [28, 37], [28, 41], [28, 42], [28, 43], [29, 30], [29, 31], [29, 32], [29, 33], [29, 34], [29, 35], [29, 41], [29, 42], [29, 43], [30, 30], [30, 31], [30, 33], [30, 34], [30, 35], [30, 41], [30, 42], [30, 43], [31, 31], [31, 32], [31, 33], [31, 34], [31, 35], [31, 36], [31, 41], [31, 42], [31, 43], [32, 31], [32, 32], [32, 34], [32, 35], [32, 36], [32, 41], [32, 42], [32, 43], [33, 41], [33, 42], [33, 43], [34, 31], [34, 41], [34, 42], [34, 43]]
Explore: [[21, 40], [23, 32], [23, 33], [23, 34], [23, 35], [23, 36], [23, 37], [23, 38], [23, 39], [24, 30], [24, 31], [24, 32], [24, 33], [24, 34], [24, 35], [24, 36], [24, 37], [24, 38], [25, 30], [25, 31], [25, 32], [25, 33], [25, 34], [25, 35], [25, 36], [25, 37], [25, 38], [26, 31], [26, 32], [26, 33], [26, 34], [26, 35], [26, 36], [26, 37], [27, 30], [27, 31], [27, 32], [27, 33], [27, 34], [27, 35], [27, 36], [27, 37], [28, 30], [28, 31], [28, 32], [28, 33], [28, 34], [28, 35], [28, 36]]
Obstacle: [[24, 29], [25, 29], [26, 29], [27, 29], [28, 29], [29, 29], [29, 36], [30, 29], [30, 36], [33, 31], [33, 32], [33, 33], [33, 34], [33, 35], [33, 36], [34, 32], [34, 33], [34, 34], [34, 35]]
Desk: [[27, 39], [27, 40], [28, 38], [28, 39], [28, 40], [29, 37], [29, 38], [29, 39], [29, 40], [30, 37], [30, 38], [31, 37], [31, 38], [32, 37], [32, 38], [33, 37], [33, 38], [33, 39], [34, 36], [34, 37], [34, 38], [34, 39], [34, 40]]
Shelf: [[30, 39], [30, 40], [31, 29], [31, 30], [31, 39], [32, 29], [32, 30], [33, 29], [33, 30], [33, 40], [34, 29], [34, 30]]
Mug: [[31, 40], [32, 39], [32, 40]]
Agent: [[32, 33]]

The movememt is forbidden on the object block.
You can only move 1 block at a time.
Your task is [['GotoLocation', ['shelf']], ['PickupObject', ['mug']], ['GotoLocation', ['desklamp']], ['ToggleObject', ['desklamp']], ['NoOp', []]]
Your subgoal is (GotoLocation, [mug])
Your position is at [32, 33]
Your available position is [[31, 33], [32, 32], [32, 34]]
Tell me what position you would move. You should first tell me your reason ,which first FIRST analyze the spatial relation between target position(if you see the subgoal object,else you should explore the room as far as you can) and your position, NEXT analyze each available position(the spatial relation between your position and it AND if it makes you near the target), and judge whether the sub-goal is finished.
ALL you OUTPUT is below, DO NOT use code block or other format:
{
  "reason": "...",
  "next_position": "(x, y)",
  "subgoal_done": "True/False"
}

**Answers:** GPT-4:
{
    "reason": "The subgoal is to go to the location of the mug. The coordinates for the mugs are at [31, 40], [32, 39], and [32, 40]. Currently, the agent is at location [32, 33]. The shortest path initially appears to be moving directly to the right from the agent's current position in order to approach one of the mug positions most directly. Moving right from [32, 33] leads to [32, 34], moving closer in the vertical lane to the target mug coordinates ([32, 39] and [32, 40]). From [32, 34], the agent can then potentially proceed further right in subsequent moves to reach [32, 39] or [32, 40]. Moving left to [32, 32] or up to [31, 33] would not progress towards the mug's location efficiently as these moves do not reduce the vertical distance to the target mug coordinates.",
    "next_position": "(32, 34)",
    "subgoal_done": "False"
}

Figure 7: GPT-4's chain-of-thought prompt and answer.

do not provide an example containing actual content, as we have found that doing so may inhibit GPT-4's ability to think independently, potentially limiting its performance.

### 6.6 OPEN-VOCABULARY PLANNING TASKS

Figure 8 depicts a step from GPT-4's open vocabulary experiment, in which it approaches the chair closest to the wall. Initially, we provide the rules and current state of the chessboard, including its size and coordinate system. Subsequently, we update GPT-4 on the chessboard's status and present the specific task and position. For this task, GPT-4 independently determines when to terminate the task. In its response, GPT-4 adopts the Chain-Of-Thought reasoning method. It accurately interprets the semantics of the chair nearest the wall and precisely identifies the coordinates. Then, it analyzes each viable move, selects two for deeper examination, and finally determines the direction for the next step.

On a 7*7 block chessboard, the rules of the game are as follows: Establish a coordinate system with the top left grid as (1,1).
Each block can be represented by coordinates. For instance, the block in the 3rd row and 5th column is denoted as (3,5).

In the chessboard, there are the following objects:In the chessboard, there are the following objects:
Desk:[[1,6],[1,7],[2,6],[2,7],[6,6],[6,7],[7,6],[7,7]]
armchair:[[2,1],[3,1],[5,7]]
sofa:[[5,1],[6,1],[7,1],[7,2],[7,3]]
table:[[3,3],[3,4],[4,3],[4,4],[5,3],[5,4]]

The movememt is forbidden on the object block. You can only move 1 block at a time.
The room is enclosed by walls. Your task is **walk around the table**!
Your position is at [2,2].
Tell me what position you will go next! If you think you don't need to move, tell me you are arrived!

**Answers: GPT-4:**
Let's break down the problem:

- You are currently at position **[2,2]**.
- The goal is to walk around the table, which occupies the following positions: **[[3,3], [3,4], [4,3], [4,4], [5,3], [5,4]]**.
- The movement is restricted by the walls (edges of the chessboard) and the objects: desks, armchairs, sofas, and the table itself.
- You can only move one block at a time.

From position **[2,2]**, you need to move toward walking around the table, which is one block below and to the right.

Now, let's consider your next move:
- You cannot move to **[3,3]** or **[3,4]** (they are table positions).
- The closest position to start walking around the table would be **[3,2]**.

Thus, your next move would be **downward**, taking you to **[3,2]**.

The movememt is forbidden on the object block. You can only move 1 block at a time.
The room is enclosed by walls. Your task is **walk around the table**!
Your position is at [3,2].
Tell me what position you will go next! If you think you don't need to move, tell me you are arrived!

**Answers: GPT-4:**
Now, you're at position **[3,2]**, and you need to continue walking around the table.

The table is occupying these positions: **[[3,3], [3,4], [4,3], [4,4], [5,3], [5,4]]**.

Your goal is to walk around the table in a counterclockwise or clockwise fashion. Since you're at **[3,2]**, you can choose to continue walking around the table.

The next step would logically be to move **downward** to **[4,2]** because that position is adjacent to the table, and it's on the correct path to continue walking around it.

So, your next move is to **[4,2]**.

The movememt is forbidden on the object block. You can only move 1 block at a time.
The room is enclosed by walls. Your task is **walk around the table**!
Your position is at [4,2].
Tell me what position you will go next! If you think you don't need to move, tell me you are arrived!

**Answers: GPT-4:**
Now you're at position **[4,2]**, and you're still walking around the table. The table occupies these positions: **[[3,3], [3,4], [4,3], [4,4], [5,3], [5,4]]**.

From **[4,2]**, the next logical step to continue walking around the table would be to move **downward** to **[5,2]**, which is also adjacent to the table and on the correct path.

So, your next move is to **[5,2]**.

The movememt is forbidden on the object block. You can only move 1 block at a time.
The room is enclosed by walls. Your task is **walk around the table**!
Your position is at [5,2].
Tell me what position you will go next! If you think you don't need to move, tell me you are arrived!

**Answers: GPT-4:**
You are currently at **[5,2]**, and you are continuing to walk around the table.

To continue walking around the table, the next logical move is to go **right** to **[6,2]**. This keeps you adjacent to the table and still on the correct path.

So, your next move is to **[6,2]**.

Figure 8: GPT-4's open-vocabulary prompt and answer-01. Task: Walk around the table.

**Answers:** GPT-4:
You're currently at [6,2], and to continue walking around the table, the next logical move is to go right to [6,3].

This keeps you adjacent to the table while continuing along the correct path.
So, your next move is to [6,3].

---

The movememt is forbidden on the object block. You can only move 1 block at a time.
The room is enclosed by walls. Your task is to **walk around the table**!
Your position is at [6,3].
Tell me what position you will go next! If you think you don't need to move, tell me you are arrived!

---

**Answers:** GPT-4:
You are currently at **[6,3]**. The table is adjacent to your north at **[5,3]**, and you want to walk around it.

From your position, the next logical move would be to move **right** to **[6,4]**, staying adjacent to the table and continuing to walk around it.

So, your next move is to **[6,4]**.

---

The movememt is forbidden on the object block. You can only move 1 block at a time.
The room is enclosed by walls. Your task is to **walk around the table**!
Your position is at [6,4].
Tell me what position you will go next! If you think you don't need to move, tell me you are arrived!

---

**Answers:** GPT-4:
You are currently at **[6,4]**, and you are continuing to walk around the table.

From this position, the next logical step to continue walking around the table would be to move **up** to **[5,4]**. However, this position is occupied by the table, so you cannot move there.

The only valid adjacent position to continue walking around the table would be to move **left** to **[6,5]**.

So, your next move is to **[6,5]**.

---

The movememt is forbidden on the object block. You can only move 1 block at a time.
The room is enclosed by walls. Your task is to **walk around the table**!
Your position is at [6,5].
Tell me what position you will go next! If you think you don't need to move, tell me you are arrived!

---

**Answers:** GPT-4:
You are currently at **[6,5]**, and you're continuing to walk around the table.

From this position, the next logical move to continue walking around the table would be to move **up** to **[5,5]**, which is adjacent to the table but not blocked by it.

So, your next move is to **[5,5]**.

---

The movememt is forbidden on the object block. You can only move 1 block at a time.
The room is enclosed by walls. Your task is to **walk around the table**!
Your position is at [5,5].
Tell me what position you will go next! If you think you don't need to move, tell me you are arrived!

---

**Answers:** GPT-4:
You are currently at **[5,5]**, and you're continuing to walk around the table.

From this position, to continue walking around the table, the next logical move is to go **up** to **[4,5]**, which keeps you adjacent to the table and continues the correct path around it.

So, your next move is to **[4,5]**.

---

The movememt is forbidden on the object block. You can only move 1 block at a time.
The room is enclosed by walls. Your task is to **walk around the table**!
Your position is at [4,5].
Tell me what position you will go next! If you think you don't need to move, tell me you are arrived!

---

**Answers:** GPT-4:
You are currently at **[4,5]**, and you are still walking around the table.

To continue walking around the table, the next logical move is to go **up** to **[3,5]**, which keeps you adjacent to the table.

So, your next move is to **[3,5]**.

Figure 9: GPT-4's open-vocabulary prompt and answer-02. Task: Walk around the table.

The movememt is forbidden on the object block. You can only move 1 block at a time.
The room is enclosed by walls. Your task is **walk around the table**!
Your position is at [3,5].
Tell me what position you will go next! If you think you don't need to move, tell me you are arrived!

**Answers:** GPT-4:
You are currently at **[3,5]**. To continue walking around the table, your next logical move is to go **right** to **[2,5]**.

So, your next move is to **[2,5]**.

The movememt is forbidden on the object block. You can only move 1 block at a time.
The room is enclosed by walls. Your task is **walk around the table**!
Your position is at [2,5].
Tell me what position you will go next! If you think you don't need to move, tell me you are arrived!

**Answers:** GPT-4:
You are currently at **[2,5]**, continuing your task to walk around the table.

The next logical move to continue walking around the table is to go **left** to **[2,4]**, which keeps you adjacent to the table and on the correct path.

So, your next move is to **[2,4]**. You haven't yet arrived, so keep moving!

The movememt is forbidden on the object block. You can only move 1 block at a time.
The room is enclosed by walls. Your task is **walk around the table**!
Your position is at [2,4].
Tell me what position you will go next! If you think you don't need to move, tell me you are arrived!

**Answers:** GPT-4:
You are currently at **[2,4]**, continuing your walk around the table.

The next logical move to continue walking around the table is to go **left** to **[2,3]**, keeping you adjacent to the table.

So, your next move is to **[2,3]**. You have arrived!

Figure 10: GPT-4's open-vocabulary promptand answer-03. Task: Walk around the table.

