# OpenReview forum: "R2C: Mapping Room to Chessboard to Unlock LLM As Low-Level Action Planner"
_ICLR.cc/2025/Conference — ICLR 2025 Conference Withdrawn Submission_

### Official Review · Reviewer_7473 · 2024-10-29

**Soundness:** 3
**Presentation:** 3
**Contribution:** 3
**Rating:** 5
**Confidence:** 4

**Summary:**

This paper presents the R2C, which projects the partially observed spatial and semantic information into a map and then uses a grid map (chessboard) to discretize the map as the unified representation for both high-level task planning and low-level action planning. This paper shows with the discrete chessboard as the environment representation, both the open-sourced large language models (Mistral 7b, LLaMA 7b) and the close-sourced large language models (GPT-4) can become efficient embodied instruction followers. To improve the decision accuracy of the LLM, this paper proposes a chain-of-thought fine-tuning framework, which asks the LLM first to answer some task-related questions and then predict the preferred action coordinates. The functions of both chessboard representation and the designed fine-tuning framework are effectively proved in the challenging ALFRED benchmarks.

**Strengths:**

(1) This paper presents an end-to-end LLM-based framework that simultaneously generates both task-level planning subgoals and low-level planning actions. This is important for the embodied instruction following agents in order to reduce the system latency and compounding errors compared with hierarchical cascaded modular approaches.

(2) This paper proposes a chain-of-thought fine-tuning method and shows that task-relevant knowledge can be effectively injected into the open-sourced LLMs (Mistral 7b, LLaMA 7b). The benchmark performance proves the fine-tuned open-sourced LLMs outperform the GPT-4 in the embodied instruction following task.

(3) The proposed chessboard representation can help generalize to open-vocabulary tasks. Experiments show the R2C can outperform many baseline methods in the ALFRED benchmark.

**Weaknesses:**

(1) Although the proposed approach is technically sound, the performance of R2C in the ALFRED is not satisfied. The listed baseline approaches are outdated. In the leaderboard of the ALFRED:https://leaderboard.allenai.org/alfred/submissions/public, there are many recent approaches that can achieve a better than 30% success rate in the unseen scenes, including both LLM-based approaches or not.
Please compare and report better approaches' performance metrics, such as EPA[1], Prompter[2], and ThinkBot[3].

(2) Some important experiments and example analyses are missing. For example, as this paper presents a chain-of-though fine-tuning for 7B LLMs, what is the zero-shot performance for LLMs without any fine-tuning? Please report the frozen LLaMA-7b and Mistral-7b performance with the same chessboard-based prompt in the ALFRED benchmark.

(3) The reference is not comprehensive, and most related works are from 2020 and 2023. Comparing with recent works can help highlight the contribution of this paper, for example, comparing with more topological-based planners, such as ConceptGraph[4], SayPlan[5], and VoroNav[6].

(4) Some typo errors should be more carefully checked. For example, on page 9, line 475 (two commas),  and page 4, line 199 (missing reference).

(5) More example visualization should be added. For example, provide detailed input prompt and output answers of fine-tuned LLaMA 7B and Mistral 7B.

Reference:

[1] Liu, X., Palacios, H. and Muise, C., 2023. Egocentric planning for scalable embodied task achievement. Advances in Neural Information Processing Systems, 36, pp.54586-54613.

[2] Inoue, Y. and Ohashi, H., 2022. Prompter: Utilizing large language model prompting for a data efficient embodied instruction following. arXiv preprint arXiv:2211.03267.

[3] Lu, G., Wang, Z., Liu, C., Lu, J. and Tang, Y., 2023. Thinkbot: Embodied instruction following with thought chain reasoning. arXiv preprint arXiv:2312.07062.

[4] Gu, Qiao, et al. "Conceptgraphs: Open-vocabulary 3d scene graphs for perception and planning." 2024 IEEE International Conference on Robotics and Automation (ICRA). IEEE, 2024.

[5] Rana, Krishan, et al. "Sayplan: Grounding large language models using 3d scene graphs for scalable task planning." CoRR (2023).

[6] Wu, Pengying, et al. "Voronav: Voronoi-based zero-shot object navigation with large language model." arXiv preprint arXiv:2401.02695 (2024).

**Questions:**

(1) What is the decision frequency of the R2C method? Does the agent decide only one grid ahead after a forward prediction from the LLMs? If that is the case, would it be rather slow in completing the entire task? Because the API-based planning approach (SayCan, Instruct2Act) only needs one LLMs' calling before finishing a long sequence of actions.

(2) In this map-based 2D decision-making problem, what are the most apparent advantages of using the dense grid coordinates as the action space rather than using landmark-based action space with an off-the-shelf path-planning algorithm?

(3) In Table 2, even if you change the perception module into the ground truth, the R2C still fails at the rate of around 60%. Can you provide more failure case analysis and discuss which potential aspects of the design can further improve the success rate?

(4) How does the dataset scale influence the fine-tuning performance in the embodied instruction following the task? And can the fine-tuned 7b LLMs still be able to maintain the original text generation abilities?

---

### Official Review · Reviewer_DPCe · 2024-10-30

**Soundness:** 2
**Presentation:** 2
**Contribution:** 2
**Rating:** 3
**Confidence:** 4

**Summary:**

This paper attempts to perform tightly coupled high-level navigation and low-level navigation in an indoor setting using LLMs. The core idea is to express low-level navigation as a grid-world game of chess, allowing the LLM to leverage world knowledge about the game to provide generalised navigation capacity for the robot. The technical approach comprises of three key steps (i) converting the raw RGB-D image data into a grid-like representation for specifying the object centric navigation task, (ii) engineering the prompt to capture the game rules, action history and the current agent-environment state and (iii) COT-D framework that guides the LLM to reason in terms of key information, direction judgement, target prediction  and selection analysis. Experimental evaluation is carried out on the Alfred data set.

**Strengths:**

- The paper concerns an important problem of leveraging LLM planning capacity at lower-levels of granularity, in a sense bridging the gap between high-level goals and the low-level motor actions.

- The central idea of relating the low-level task with a game-like representation with the aim of generalization is interesting.

- The investigation into encoding the raw sensor input into the task setting is realistic and insightful.

**Weaknesses:**

The primary concern is how general and automated is the process of converting the navigation task into a game. The paper makes one ponder what class of navigation tasks can be encoded in which class of games. The paper provides some indicated results with the chess game but the generality of the result and the encoding process is not clear. The authors are encouraged to provide specific examples of different types of navigation tasks and discuss how (or if) they could be encoded using their Room to Chessboard framework. Further, the authors are requested to include a more detailed discussion of the limitations of their approach in terms of task types that may not be easily represented as a chess-like game.

**Questions:**

1. How general is the process of encoding the navigation task into a chess game? Is it confined to object centric navigation or does it extend to re-arrangement planning, task and motion planning problems (where the agent may need to push away objects to make space). For example, https://www.ijcai.org/proceedings/2018/0674.pdf and https://people.cs.rutgers.edu/~kb572/pubs/fast_object_rearrangement.pdf for examples of such tasks. Please indicate if the framework can or cannot solve such planning tasks.

2. In case an action fails to execute, does the system re-plan?

3. The authors mention related works such as LM-Nav. If feasible, a comparison with such an approach would be insightful.

4. Is it feasible to extend your approach to incorporating robot actions that are not present in a chess game. For example, consider the "jumping onto an object" action for a quadruped robot since Chess does not allow pieces to be in the same cell. Similarly, if we have an object transport task where small objects are kept onto a tray and carried from location to location, would the system be able to perform such reasoning. If feasible, authors are requested to discuss specific modifications they might make to their framework to accommodate these types of actions, or to explain why such extensions might be challenging within their current approach.

---

### Official Review · Reviewer_sWm1 · 2024-11-04

**Soundness:** 3
**Presentation:** 3
**Contribution:** 2
**Rating:** 6
**Confidence:** 4

**Summary:**

This paper proposes a novel framework to enable LLMs to perform low-level action planning for robotic tasks. Traditional LLM applications in robotics focus on high-level task planning, while low-level actions rely on other specialized controllers. This paper addresses the challenge of communication between the LLM and the robot by mapping a room environment into a chessboard-style grid, called Room to Chessboard (R2C). This grid provides a simplified, yet semantically rich representation of the environment that LLMs can use to generate precise, step-by-step navigation instructions.
To further enhance the LLM's decision-making, the authors introduce a Chain-of-Thought Decision (CoT-D) fine-tuning paradigm that strengthens spatial reasoning and interpretability. Tested on the ALFRED benchmark, R2C outperforms other LLM-based methods and achieves competitive results with specialist models in complex, long-horizon tasks. The paper demonstrates that R2C can generalize across seen and unseen environments and handle open-vocabulary tasks, underscoring its versatility and scalability.

**Strengths:**

The R2C framework's chessboard representation provides a concise yet effective way for LLMs to interpret spatial information, bridging the communication gap between high-level instructions and low-level execution. The Chain-of-Thought Decision (CoT-D) fine-tuning paradigm adds explainability and improves spatial reasoning, allowing LLMs to handle complex decision-making in low-level action planning.
The framework demonstrates state-of-the-art results among LLM-based methods and competitive results compared to specialist models, showcasing its efficacy in both seen and unseen environments. Additionally, it extends beyond traditional benchmarks by enabling LLMs to perform well in open-ended tasks, which are closer to real-world scenarios

**Weaknesses:**

Although the paper compares R2C with LLM-based baselines, a more detailed comparison with non-LLM approaches, such as reinforcement learning or specialized motion planners, would better position R2C within the robotics field. Experiments are limited to simulated environments (ALFRED benchmark), and it remains unclear how the R2C framework would perform in real-world settings with more complex and dynamic elements.

**Questions:**

The authors can consider comparing R2C with non-LLM specialist models. They can also address the increased computational cost introduced by CoT-D fine-tuning for real-time deployment and find if there are ways to streamline or selectively apply CoT-D only when necessary. How adaptable R2C is to different types of interaction tasks beyond navigation might be questionable.

---

### Official Review · Reviewer_uZMP · 2024-11-04

**Soundness:** 3
**Presentation:** 3
**Contribution:** 1
**Rating:** 3
**Confidence:** 4

**Summary:**

## Research Question
Though possess extensive world knowledge and demonstrate generalizationability, LLMs lack the spatial awareness of real-world environments. On one hand, the Robot can hardly convey spatial information from the environment to the LLM. On the other hand, the LLM struggles to efficiently communicate low-level decisions to the Robot.
## Method
To this end, the author proposed Room to Chessboard (R2C) framework to establish a “common language” between LLM and the Robot. Such a framework will unlock LLM as low-level action planner and develop an explainable chain of thought decision analysis paradigm. More specifically, a task will first be translated into sub-goals as high-level planning. Then R2C will translate the task-aware environment information into a compact chessboard, therefore LLM can perform low-level planning on the chessboard by predicting the robot’s next position.  Additionally, the author formalized a Chain of Thought Decision (CoT-D) task for LLM to enhance its overall spatial reasoning, and designed a fine-tuning paradigm corresponding to the CoT-D tasks.

**Strengths:**

1. A practical research question and reasonable insight

The research question proposed by the authors, i.e. "LLMs lack the spatial awareness of real-world environments", is exactly one of the major bottlenecks for LLM-based robotic planning, many works, including this paper, are trying to mitigate the mismatch between planning space in LLM and planning space in real world.  Therefore, the insight of this paper: establishing an efficient communication interface between LLMs and robots is very natural and reasonable.

2. Extensive experiments

The authors conducted comparisons between different LLMs(GPT-4, fine-tuned Llama, fine-tuned Mistral) with an additional exploration on open-vocabulary tasks.

**Weaknesses:**

1. The problem setting is too simple.

1.1 As the final goal is to let LLM do the low-level planning, a 2D grid world setting is far from practical or useful,  low-level planning, e.g. robot arm manipulation, in a 3D world is far more complicated compared with a 2D grid world.

1.2 Another concern is whether LLM outperforms traditional planning algorithms (e.g. Dijkstra, A star) in step-by-step navigation tasks, e.g. a simple baseline could be an LLM-specified target position + heuristic planner.

2. Weak baseline and unstable improvement

The selection of baseline is questionable since 1)some heuristic planners (e.g. A*) are also applicable to such task(navigation on 2D map), could the authors compare their approach to heuristic planners like A* for the 2D navigation tasks? ; 2) Saycan is classic enough as a baseline, can the authors provide details on their implementation of SayCan, particularly how they handled the reinforcement learning component? 3) Specific to the LLM-based high-level planning, there are more works published with code in the 2023~2024 period(e.g. Auto-TAMP, code as policy, etc..), but this is just a minor point. 4) The performance improvement according to Table 2 is not stable enough.

3. Insufficient impact on communities

One question for the paper's impact could be:  "How do the authors envision their approach evolving as language model capabilities improve?"  or  "Are there aspects of the R2C framework beyond prompt engineering that would remain relevant even with more advanced models?"

**Questions:**

Dear authors:

Generally, the paper's idea is doable and insightful, but the solution details can be of great improvement.

 I will certainly raise my point if my concerns below are mitigated:
1. Have the authors considered extending their approach to 3D action spaces? What challenges do you anticipate in such an extension?
2. The concern of "Weak baseline and unstable improvement" in Weakness.

---

### Note · Authors · 2024-11-15

**Comment:**

We sincerely thank the Area Chair and all the reviewers for their valuable feedback and constructive suggestions. After careful consideration, we have decided to further improve our work and withdraw this version of the submission.

**Withdrawal Confirmation:**

I have read and agree with the venue's withdrawal policy on behalf of myself and my co-authors.